# Adaptive Fixed-Time Neural Networks Control for Pure-Feedback Non-Affine Nonlinear Systems with State Constraints

**DOI:** 10.3390/e24050737

**Published:** 2022-05-22

**Authors:** Yang Li, Quanmin Zhu, Jianhua Zhang, Zhaopeng Deng

**Affiliations:** 1School of Information and Control Engineering, Qingdao University of Technology, Qingdao 266525, China; yang_li@qut.edu.cn (Y.L.); dengzhaopeng@qut.edu.cn (Z.D.); 2Department of Engineering Design and Mathematics, University of the West of England, Coldharbour Lane, Bristol BS16 1QY, UK; quan.zhu@uwe.ac.uk

**Keywords:** adaptive control, neural network control, nonlinear constraint systems, non-affine nonlinear systems, pure feedback

## Abstract

A new fixed-time adaptive neural network control strategy is designed for pure-feedback non-affine nonlinear systems with state constraints according to the feedback signal of the error system. Based on the adaptive backstepping technology, the Lyapunov function is designed for each subsystem. The neural network is used to identify the unknown parameters of the system in a fixed-time, and the designed control strategy makes the output signal of the system track the expected signal in a fixed-time. Through the stability analysis, it is proved that the tracking error converges in a fixed-time, and the design of the upper bound of the setting time of the error system only needs to modify the parameters and adaptive law of the controlled system controller, which does not depend on the initial conditions.

## 1. Introduction

In recent years, great breakthroughs have been made in the research of adaptive trajectory tracking control for uncertain nonlinear systems [1,2,3]. When solving such problems, neural network technology has become the key technology [4,5,6]. Combining neural network technology with backstepping control and adaptive control, the results have been widely used in different types of nonlinear systems such as strict feedback and pure feedback [7,8,9]. With the development of increasing power integrators, great progress has been made in the research of non-affine nonlinear systems. In recent years, the problems studied include output feedback stability, state output constraints, etc. Many methods have been introduced to solve these problems, such as backstepping technology, adaptive technology, and neural network control [10,11,12]. For nonlinear systems with time delays, the authors of reference [13] designed the control strategy by combining adaptive neural network and backstepping technology, and then the neural network technology based on adaptive backstepping was developed and applied [14,15,16].

With the development of society, the accuracy requirements of industrial control systems for convergence time are increasing. For example, in antimissile control systems, aircraft attitude control systems, and robot control systems, the purpose of controller design is to realize the stability of the controlled system and maintain stability in finite time (for example, in antimissile control systems, there is no need for control after missile explosion). For nonlinear systems with uncertainties, researchers have combined fixed-time controls with adaptive neural network technology to produce many excellent control schemes [17,18,19].

Researchers combine neural networks with adaptive control for online identification of complex nonlinear objects. In the design of these control systems, neural networks are generally used to approximate the uncertain nonlinear terms of the system, and neural networks are effective in compact sets [20,21,22]. In recent years, some fixed-time control methods based on the neural networks have been developed [23,24,25]. The author of reference [26] studies the control method of unknown nonlinear systems based on free model control. Based on Lyapunov functional and analysis technology, combined with advanced control algorithms, sufficient conditions for the master–slave memristor systems to realize timing synchronization are established. The authors of reference [27] extended the method to time-varying delay discontinuous fuzzy inertial neural network fixed-time synchronous control.

Although the research on fixed-time adaptive neural network control has produced a series of research results, there are still many problems to be solved in the existing control strategies, such as system constraints. In reference [28,29,30], the control problem of constrained nonlinear systems is discussed. For the control problem of systems with state constraints, the difficulty of constraints can be solved by using the boundary Lyapunov function. However, for the control problem of constrained non-affine nonlinear systems, the control strategies in the above literature cannot be used directly, and the research results based on fixed-time control are relatively few.

In summary, when there are state constraints in non-affine nonlinear systems, how to combine the adaptive neural network control and backstepping control to design effective control strategies so that the system can achieve the expected performance in fixed time, with the setting time not depending on the initial state of the system, is a problem. To solve this problem, some control problems have not been solved, such as for pure-feedback non-affine nonlinear systems, how to combine the backstepping method with Lyapunov function theory to design a fixed-time adaptive neural network tracking control strategy, so that the system output can track the desired signal and maintain fixed-time stability, the control performance can be guaranteed without initial conditions, and all state variables are bounded to a fixed region.

This article consists of the following parts. In Section 2, a constrained nonlinear system mathematical description of the problem is presented. In Section 3, firstly, the novel fixed-time stability theorem for constrained nonlinear systems is proposed, secondly, the adaptive neural network fixed-time tracking control scheme for constrained nonlinear systems is presented. In Section 4, the performances of the tracking control scheme are illustrated by a simulation example. In Section 5, some conclusions of the article are summarized.

## 2. Problem Formation and Preliminaries

Based on backstepping technology, combined with an adaptive neural network and fixed-time control, the tracking control of pure-feedback non-affine nonlinear interconnected systems was studied. Consider pure-feedback nonlinear systems:(1)$$\left\{\begin{array}{l} {\dot{x}}_{i}\left(t\right)=f_{i}\left({\bar{x}}_{i+1}\left(t\right)\right),i=1,2,\ldots ,n-1\\ {\dot{x}}_{n}\left(t\right)=f_{n}\left({\bar{x}}_{n}\left(t\right),u\left(t\right)\right)\\ y\left(t\right)=x_{1}\left(t\right) \end{array}\right.$$
where $x={\left(\begin{array}{cccc} x_{1} & x_{2} & \cdots & x_{n} \end{array}\right)}^{T}\in {ℜ}^{n}$, u∈ℜ, y∈ℜ, indicate the state, control and output, respectively, $f_{i}\left(\cdot \right),i=1,2,\ldots ,n$ are nonlinear smooth functions, $y_{d}\in ℜ$ is desired trajectory.

**Remark** **1.**
*Based on the existing algorithms, this article attempts to further design a novel neural network adaptive control algorithm. The control objective of the algorithm is the output of the pure-feedback non-affine nonlinear system that can track the desired signal and maintain fixed-time stability. The designed upper bound of the setting time does not rely on the initial parameters, only by adjusting the parameters of the controller.*


**Lemma** **1**[6]**.**
*For*
$x_{i}\in R$
*and*
$x_{i}\geq 0$*,*
i=1,2,⋯,n*,*
0<p<1*,*
q>1
*, then*
(2)$${\left(\sum_{i=1}^{n}x_{i}\right)}^{p}\leq \sum_{i=1}^{n}x_{i}{}^{p}\leq n^{1-p}{\left(\sum_{i=1}^{n}x_{i}\right)}^{p}$$
(3)$$n^{1-q}{\left(\sum_{i=1}^{n}x_{i}\right)}^{q}\leq \sum_{i=1}^{n}x_{i}{}^{q}\leq {\left(\sum_{i=1}^{n}x_{i}\right)}^{q}$$

## 3. Main Results

The control algorithm was designed for the system (1). The objective of the control was to propose a new adaptive fixed-time neural network tracking control algorithm for the pure-feedback nonlinear system. Adaptive neural network technology is used to solve the uncertainty of the unknown system. Under the proposed control scheme, through the Lyapunov stability analysis, the closed system is fixed-time stability.

For a nonlinear system (1), combine homeomorphism mapping and backstepping control to design constraint control, in the first step, consider system state
(4)$$z_{1}=x_{1}-y_{d}$$

Design homeomorphism mapping
(5)$$\xi_{1}=\mathrm{arctanh}\left(\frac{z_{1}}{k_{b1}}\right)$$
where $k_{b1}>0$ is the bound of $z_{1}$ and satisfy the $\left|z_{1}\right|<k_{b1}$, then the system can obtain
(6)$$\begin{array}{c} {\dot{\xi }}_{1}=\frac{k_{b1}}{k_{b1}^{2}-z_{1}^{2}}{\dot{z}}_{1}\\ \;\;\;\;\;\;\;=\frac{k_{b1}}{k_{b1}^{2}-z_{1}^{2}}\left(f_{1}\left(x_{1},x_{2}\right)-{\dot{y}}_{d}\right) \end{array}$$

Choose the NN to approximate the nonlinear system $f_{1}\left({\bar{x}}_{2}\right)$,${\bar{x}}_{2}\in \Omega_{1}\subset {ℜ}^{2}$ and $\Omega_{1}$ is compact set
(7)$$\frac{k_{b1}}{k_{b1}^{2}-z_{1}^{2}}\left(f_{1}\left(x_{1},x_{2}\right)-{\dot{y}}_{d}\right)=W_{1}^{T*}\Psi_{1}\left({\bar{x}}_{2}\right)+\varepsilon_{1}$$
where $\left\|W_{1}^{*}\right\|=\theta_{1}$, ${\hat{\theta }}_{1}$ is estimation of $\theta_{1}$ and ${\tilde{\theta }}_{1}={\hat{\theta }}_{1}-\theta_{1}$, then we have
(8)$$W_{1}^{*T}\Psi_{1}\left({\bar{x}}_{2}\right)\leq \theta_{1}\left\|\Psi_{1}\left({\bar{x}}_{2}\right)\right\|$$

Define a Lyapunov functional candidate as
(9)$$V_{1}=\frac{1}{2}\xi_{1}^{2}+\frac{1}{2\mu_{1}}{\tilde{\theta }}_{1}^{2}$$
take the time derivative (9) along the trajectory of (6) as
(10)$$\begin{array}{c} {\dot{V}}_{1}=\xi_{1}\frac{k_{b1}}{k_{b1}^{2}-z_{1}^{2}}\left(f_{1}\left(x_{1},x_{2}\right)-{\dot{y}}_{d}\right)+\frac{1}{\mu_{1}}{\tilde{\theta }}_{1}{\dot{\hat{\theta }}}_{1}\\ =\xi_{1}\left(W_{1}^{*T}\Psi_{1}\left({\bar{x}}_{2}\right)+\varepsilon_{1}\right)+\frac{1}{\mu_{1}}{\tilde{\theta }}_{1}{\dot{\hat{\theta }}}_{1} \end{array}$$

Choose the virtual control law
(11)$$\xi_{2}=k_{1}\xi_{1}+k_{p1}\xi_{1}^{p}+k_{q1}\xi_{1}^{q}+sign\left(\xi_{1}\right){\hat{\theta }}_{1}\left\|\Psi_{1}\left({\bar{x}}_{2}\right)\right\|$$
where $k_{1}>\frac{1}{2},k_{p1}>0,k_{q1}>0,0<p<1,q>1$, based on homeomorphism mapping
(12)$$z_{2}=k_{b2}\tanh\left(\xi_{2}\right)$$
where $k_{b2}>0$ is the bound of $z_{2}$ and satisfies the $\left|z_{2}\right|<k_{b2}$, and
(13)$$\alpha_{1}=x_{2}-z_{2}$$
then we have
(14)$$\begin{array}{cc} {\dot{V}}_{1} & =\xi_{1}W_{1}^{*T}\Psi_{1}\left({\bar{x}}_{2}\right)+\xi_{1}\varepsilon_{1}-k_{1}\xi_{1}^{2}-k_{p1}\xi_{1}^{p+1}-k_{q1}\xi_{1}^{q+1}\\ & -{\hat{\theta }}_{1}\left|\xi_{1}\right|\left\|\Psi_{1}\left({\bar{x}}_{2}\right)\right\|+\xi_{1}\xi_{2}+\frac{1}{\mu_{1}}{\tilde{\theta }}_{1}{\dot{\hat{\theta }}}_{1} \end{array}$$
when $\left\|W_{1}^{*}\right\|=\theta_{1}$, we have
(15)$$\xi_{1}W_{1}^{*T}\Psi_{1}\left({\bar{x}}_{2}\right)\leq \theta_{1}\left|\xi_{1}\right|\left\|\Psi_{1}\left({\bar{x}}_{2}\right)\right\|$$
and
(16)$$\xi_{1}\varepsilon_{1}\leq \frac{1}{2}\xi_{1}^{2}+\frac{1}{2}\varepsilon_{1}^{2}$$
then we have
(17)$$\begin{array}{cc} {\dot{V}}_{1} & =-{\tilde{\theta }}_{1}\left|\xi_{1}\right|\left\|\Psi_{1}\left({\bar{x}}_{2}\right)\right\|+\frac{1}{2}\xi_{1}^{2}+\frac{1}{2}\varepsilon_{1}^{2}-k_{1}\xi_{1}^{2}-k_{p1}\xi_{1}^{p+1}-k_{q1}\xi_{1}^{q+1}\\ & +\xi_{1}\xi_{2}+\frac{1}{\mu_{1}}{\tilde{\theta }}_{1}{\dot{\hat{\theta }}}_{1} \end{array}$$
where
(18)$${\tilde{\theta }}_{1}={\hat{\theta }}_{1}-\theta_{1}$$

Choose the NN adaptive law as
(19)$${\dot{\hat{\theta }}}_{1}=\mu_{1}\left(\left|\xi_{1}\right|\left\|\Psi_{1}\left({\bar{x}}_{2}\right)\right\|-\rho_{p1}{\hat{\theta }}_{1}^{p}-\rho_{q1}{\hat{\theta }}_{1}^{q}\right),{\hat{\theta }}_{1}\left(0\right)=0$$
where $\mu_{1}>0,\rho_{p1}>0,\rho_{q1}>0$, then we have
(20)$$\begin{array}{cc} {\dot{V}}_{1} & =-{\tilde{\theta }}_{1}\left|\xi_{1}\right|\left\|\Psi_{1}\left({\bar{x}}_{2}\right)\right\|+\frac{1}{2}\xi_{1}^{2}+\frac{1}{2}\varepsilon_{1}^{2}-k_{1}\xi_{1}^{2}-k_{p1}\xi_{1}^{p+1}-k_{q1}\xi_{1}^{q+1}\\ & +\xi_{1}\xi_{2}+{\tilde{\theta }}_{1}\left(\left|\xi_{1}\right|\left\|\Psi_{1}\left({\bar{x}}_{2}\right)\right\|-\rho_{p1}{\hat{\theta }}_{1}^{p}-\rho_{q1}{\hat{\theta }}_{1}^{q}\right)\\ & =-\left(k_{1}-\frac{1}{2}\right)\xi_{1}^{2}-k_{p1}\xi_{1}^{p+1}-k_{q1}\xi_{1}^{q+1}+\xi_{1}\xi_{2}\\ & -\rho_{p1}{\tilde{\theta }}_{1}{\hat{\theta }}_{1}^{p}-\rho_{q1}{\tilde{\theta }}_{1}{\hat{\theta }}_{1}^{q}+\frac{1}{2}\varepsilon_{1}^{2} \end{array}$$
based on inequalities from [7], the following hold:(21)$$\begin{array}{l} -\rho_{p1}{\tilde{\theta }}_{1}{\hat{\theta }}_{1}^{p}\leq -\varsigma_{p1}{\tilde{\theta }}_{1}^{p+1}+\upsilon_{p1}\theta_{1}^{p+1}\\ -\rho_{q1}{\tilde{\theta }}_{1}{\hat{\theta }}_{1}^{q}\leq -\varsigma_{q1}{\tilde{\theta }}_{1}^{q+1}+\upsilon_{q1}\theta_{1}^{q+1} \end{array}$$
where $\rho_{p1},\varsigma_{p1},\upsilon_{p1},\rho_{q1},\varsigma_{q1},\upsilon_{q1}>0$, therefore, we have
(22)$$\begin{array}{cc} {\dot{V}}_{1} & \leq -\left(k_{1}-\frac{1}{2}\right)\xi_{1}^{2}-k_{p1}\xi_{1}^{p+1}-k_{q1}\xi_{1}^{q+1}+\xi_{1}\xi_{2}-\varsigma_{p1}{\tilde{\theta }}_{1}^{p+1}+\upsilon_{p1}\theta_{1}^{p+1}\\ & -\varsigma_{q1}{\tilde{\theta }}_{1}^{q+1}+\upsilon_{q1}\theta_{1}^{q+1}+\frac{1}{2}\varepsilon_{1}^{2}\\ & \leq -k_{p1}\xi_{1}^{p+1}-k_{q1}\xi_{1}^{q+1}-\varsigma_{p1}{\tilde{\theta }}_{1}^{p+1}-\varsigma_{q1}{\tilde{\theta }}_{1}^{q+1}+\xi_{1}\xi_{2}+\delta_{1} \end{array}$$
where
(23)$$\delta_{1}=\frac{1}{2}\varepsilon_{1}^{2}+\upsilon_{p1}\theta_{1}^{p+1}+\upsilon_{q1}\theta_{1}^{q+1}$$
The ith step 2≤i≤n, consider system state
(24)$$z_{i}=x_{i}-\alpha_{i-1}$$
Design homeomorphism mapping
(25)$$\xi_{i}=\mathrm{arctanh}\left(\frac{z_{i}}{k_{bi}}\right)$$
where $k_{bi}>0$ is the bound of $z_{i}$ and satisfies the $\left|z_{i}\right|<k_{bi}$, then the system can obtain
(26)$$\begin{array}{cc} {\dot{\xi }}_{i} & =\frac{k_{bi}}{k_{bi}^{2}-z_{i}^{2}}{\dot{z}}_{i}\\ & =\frac{k_{bi}}{k_{bi}^{2}-z_{i}^{2}}\left(f_{i}-{\dot{\alpha }}_{i-1}\right) \end{array}$$

The neural network is constructed as $f_{i}$,${\bar{x}}_{i+1}\in \Omega_{i}\subset {ℜ}^{i+1}$ and $\Omega_{i}$ is compact set
(27)$$\frac{k_{bi}}{k_{bi}^{2}-z_{i}^{2}}\left(f_{i}-{\dot{\alpha }}_{i-1}\right)=W_{i}^{T*}\Psi_{i}\left({\bar{x}}_{i+1}\right)+\varepsilon_{i}$$
where $\left\|W_{i}^{*}\right\|=\theta_{i}$, ${\hat{\theta }}_{i}$ is estimation of $\theta_{i}$ and ${\tilde{\theta }}_{i}={\hat{\theta }}_{i}-\theta_{i}$, then we have
(28)$$W_{i}^{*T}\Psi_{1}\left({\bar{x}}_{i+1}\right)\leq \theta_{i}\left\|\Psi_{i}\left({\bar{x}}_{i+1}\right)\right\|$$

Define a Lyapunov functional candidate as
(29)$$V_{i}=\frac{1}{2}\xi_{i}^{2}+\frac{1}{2\mu_{i}}{\tilde{\theta }}_{i}^{2}$$

Take the time derivative (29) along the trajectory of (26) as
(30)$$\begin{array}{cc} {\dot{V}}_{i} & =\xi_{i}\frac{k_{bi}}{1-z_{i}^{2}}\left(f_{i}-{\dot{\alpha }}_{i-1}\right)+\frac{1}{\mu_{i}}{\tilde{\theta }}_{i}{\dot{\hat{\theta }}}_{i}\\ & =\xi_{i}\left(W_{i}^{*T}\Psi_{i}\left({\bar{x}}_{i+1}\right)+\varepsilon_{i}\right)+\frac{1}{\mu_{i}}{\tilde{\theta }}_{i}{\dot{\hat{\theta }}}_{i} \end{array}$$

The virtual control signal is constructed as
(31)$$\xi_{i+1}=\xi_{i-1}+k_{i}\xi_{i}+k_{pi}\xi_{i}^{p}+k_{qi}\xi_{i}^{q}+sign\left(\xi_{i}\right){\hat{\theta }}_{i}\left\|\Psi_{i}\left({\bar{x}}_{i+1}\right)\right\|$$
where $k_{i}>\frac{1}{2},k_{pi}>0,k_{qi}>0,0<p<1,q>1$, based on homeomorphism mapping
(32)$$z_{i+1}=k_{bi+1}\tanh\left(\xi_{i+1}\right)$$
where $k_{bi+1}>0$ is the bound of $z_{i+1}$ and satisfies the $\left|z_{i+1}\right|<k_{bi+1}$ where
(33)$$z_{i+1}=x_{i+1}-\alpha_{i}$$
and assume $x_{n+1}=u$, then we have
(34)$$\begin{array}{cc} {\dot{V}}_{i} & =\xi_{i}W_{i}^{*T}\Psi_{i}\left({\bar{x}}_{i+1}\right)+\xi_{i}\varepsilon_{i}-\xi_{i-1}\xi_{i}-k_{i}\xi_{i}^{2}-k_{pi}\xi_{i}^{p+1}-k_{qi}\xi_{i}^{q+1}\\ & -{\hat{\theta }}_{i}\left|\xi_{i}\right|\left\|\Psi_{i}\left({\bar{x}}_{i+1}\right)\right\|+\xi_{i}\xi_{i+1}+\frac{1}{\mu_{i}}{\tilde{\theta }}_{i}{\dot{\hat{\theta }}}_{i} \end{array}$$
when $\left\|W_{i}^{*}\right\|=\theta_{i}$, we have
(35)$$\xi_{i}W_{i}^{*T}\Psi_{i}\left({\bar{x}}_{i+1}\right)\leq \theta_{i}\left|\xi_{i}\right|\left\|\Psi_{i}\left({\bar{x}}_{i+1}\right)\right\|$$
and
(36)$$\xi_{i}\varepsilon_{i}\leq \frac{1}{2}\xi_{i}^{2}+\frac{1}{2}\varepsilon_{i}^{2}$$
then we have
(37)$$\begin{array}{cc} {\dot{V}}_{i} & =-{\tilde{\theta }}_{i}\left|\xi_{i}\right|\left\|\Psi_{i}\left({\bar{x}}_{i+1}\right)\right\|+\frac{1}{2}\xi_{i}^{2}+\frac{1}{2}\varepsilon_{i}^{2}-k_{1}\xi_{i}^{2}-k_{pi}\xi_{i}^{p+1}-k_{qi}\xi_{i}^{q+1}\\ & -\xi_{i-1}\xi_{i}+\xi_{i}\xi_{i+1}+\frac{1}{\mu_{i}}{\tilde{\theta }}_{i}{\dot{\hat{\theta }}}_{i} \end{array}$$
where
(38)$${\tilde{\theta }}_{i}={\hat{\theta }}_{i}-\theta_{i}$$

The NN adaptive signal is constructed as
(39)$${\dot{\hat{\theta }}}_{i}=\mu_{i}\left(\left|\xi_{i}\right|\left\|\Psi_{i}\left({\bar{x}}_{i+1}\right)\right\|-\rho_{pi}{\hat{\theta }}_{i}^{p}-\rho_{qi}{\hat{\theta }}_{i}^{q}\right),{\hat{\theta }}_{i}\left(0\right)=0$$
where $\mu_{i}>0,\rho_{pi}>0,\rho_{qi}>0$, then we have
(40)$$\begin{array}{cc} {\dot{V}}_{i} & =-{\tilde{\theta }}_{i}\left|\xi_{i}\right|\left\|\Psi_{i}\left({\bar{x}}_{i+1}\right)\right\|+\frac{1}{2}\xi_{i}^{2}+\frac{1}{2}\varepsilon_{i}^{2}-k_{i}\xi_{i}^{2}-k_{pi}\xi_{i}^{p+1}-k_{qi}\xi_{i}^{q+1}\\ & -\xi_{i-1}\xi_{i}+\xi_{i}\xi_{i+1}+{\tilde{\theta }}_{i}\left(\left|\xi_{i}\right|\left\|\Psi_{i}\left({\bar{x}}_{i+1}\right)\right\|-\rho_{pi}{\hat{\theta }}_{i}^{p}-\rho_{qi}{\hat{\theta }}_{i}^{q}\right)\\ & =-\left(k_{i}-\frac{1}{2}\right)\xi_{i}^{2}-k_{pi}\xi_{i}^{p+1}-k_{qi}\xi_{i}^{q+1}-\xi_{i-1}\xi_{i}+\xi_{i}\xi_{i+1}\\ & -\rho_{pi}{\tilde{\theta }}_{i}{\hat{\theta }}_{i}^{p}-\rho_{qi}{\tilde{\theta }}_{i}{\hat{\theta }}_{i}^{q}+\frac{1}{2}\varepsilon_{i}^{2} \end{array}$$
Based on inequalities from [7], the following hold:(41)$$\begin{array}{l} -\rho_{pi}{\tilde{\theta }}_{i}{\hat{\theta }}_{i}^{p}\leq -\varsigma_{pi}{\tilde{\theta }}_{i}^{p+1}+\upsilon_{pi}\theta_{i}^{p+1}\\ -\rho_{qi}{\tilde{\theta }}_{i}{\hat{\theta }}_{i}^{q}\leq -\varsigma_{qi}{\tilde{\theta }}_{i}^{q+1}+\upsilon_{qi}\theta_{i}^{q+1} \end{array}$$
where $\rho_{pi},\varsigma_{pi},\upsilon_{pi},\rho_{qi},\varsigma_{qi},\upsilon_{qi}>0$, therefore we have
(42)$$\begin{array}{cc} {\dot{V}}_{i} & \leq -\left(k_{i}-\frac{1}{2}\right)\xi_{i}^{2}-k_{pi}\xi_{i}^{p+1}-k_{qi}\xi_{i}^{q+1}-\xi_{i-1}\xi_{i}+\xi_{i}\xi_{i+1}-\varsigma_{pi}{\tilde{\theta }}_{i}^{p+1}\\ & +\upsilon_{pi}\theta_{i}^{p+1}-\varsigma_{qi}{\tilde{\theta }}_{i}^{q+1}+\upsilon_{qi}\theta_{i}^{q+1}+\frac{1}{2}\varepsilon_{i}^{2}\\ & \leq -\xi_{i-1}\xi_{i}-k_{pi}\xi_{i}^{p+1}-k_{qi}\xi_{i}^{q+1}-\varsigma_{pi}{\tilde{\theta }}_{i}^{p+1}-\varsigma_{qi}{\tilde{\theta }}_{i}^{q+1}+\xi_{i}\xi_{i+1}+\delta_{i} \end{array}$$
where
(43)$$\delta_{i}=\frac{1}{2}\varepsilon_{i}^{2}+\upsilon_{pi}\theta_{i}^{p+1}+\upsilon_{qi}\theta_{i}^{q+1}$$
The n+1th step, this is the most important step.
(44)$$z_{n+1}=u-\alpha_{n}$$
Based on system
(45)$${\dot{z}}_{n+1}=v-{\dot{\alpha }}_{n}$$
Design homeomorphism mapping
(46)$$\xi_{n+1}=\mathrm{arctanh}\left(\frac{z_{n+1}}{k_{bn+1}}\right)$$
where $k_{bn+1}>0$ is the bound of $z_{n+1}$ and satisfies the $\left|z_{n+1}\right|<k_{bn+1}$, then the system can obtain
(47)$$\begin{array}{cc} {\dot{\xi }}_{n+1} & =\frac{k_{bn+1}}{k_{bn+1}^{2}-z_{n+1}^{2}}{\dot{z}}_{n+1}\\ & =\frac{k_{bn+1}}{k_{bn+1}^{2}-z_{n+1}^{2}}\left(v-{\dot{\alpha }}_{n}\right) \end{array}$$
The neural network is constructed as ${\dot{\alpha }}_{n}$,${\bar{x}}_{n+1}=\left(x,u\right)\in \Omega_{n+1}\subset {ℜ}^{n+1}$ and $\Omega_{n+1}$ is compact set
(48)$$\frac{k_{bn+1}}{k_{bn+1}^{2}-z_{n+1}^{2}}{\dot{\alpha }}_{n}=W_{n+1}^{T*}\Psi_{n+1}\left({\bar{x}}_{n+1}\right)+\varepsilon_{n+1}$$
where $\left\|W_{n+1}^{*}\right\|=\theta_{n+1}$, ${\hat{\theta }}_{n+1}$ is an estimation of $\theta_{n+1}$ and ${\tilde{\theta }}_{n+1}={\hat{\theta }}_{n+1}-\theta_{n+1}$, then we have
(49)$$W_{n+1}^{*T}\Psi_{n+1}\left({\bar{x}}_{n+1}\right)\leq \theta_{n+1}\left\|\Psi_{n+1}\left({\bar{x}}_{n+1}\right)\right\|$$
Define a Lyapunov functional candidate as
(50)$$V_{n+1}=\frac{1}{2}\xi_{n+1}^{2}+\frac{1}{2\mu_{n+1}}{\tilde{\theta }}_{n+1}^{2}$$
Take the time derivative (9) along the trajectory of (6) as
(51)$$\begin{array}{cc} {\dot{V}}_{n+1} & =\xi_{n+1}\frac{k_{bn+1}}{k_{bn+1}^{2}-z_{n+1}^{2}}\left(v-{\dot{\alpha }}_{i}\right)+\frac{1}{\mu_{n+1}}{\tilde{\theta }}_{n+1}{\dot{\hat{\theta }}}_{n+1}\\ & =\xi_{n+1}\frac{k_{bn+1}}{k_{bn+1}^{2}-z_{n+1}^{2}}v-\xi_{n+1}\left(W_{n+1}^{*T}\Psi_{n+1}\left({\bar{x}}_{n+1}\right)+\varepsilon_{n+1}\right)\\ & +\frac{1}{\mu_{n+1}}{\tilde{\theta }}_{n+1}{\dot{\hat{\theta }}}_{n+1} \end{array}$$
Choose the control
(52)$$v=\frac{-\xi_{n}-k_{n+1}\xi_{n+1}-k_{pn+1}\xi_{n+1}^{p}-k_{qn+1}\xi_{n+1}^{q}-sign\left(\xi_{n+1}\right){\hat{\theta }}_{n+1}\left\|\Psi_{n+1}\left({\bar{x}}_{n+1}\right)\right\|}{\frac{k_{bn+1}}{k_{bn+1}^{2}-z_{n+1}^{2}}}$$
where $k_{n+1}>\frac{1}{2},k_{pn+1}>0,k_{qn+1}>0,0<p<1,q>1$, then
(53)$$\begin{array}{cc} {\dot{V}}_{n+1} & =-\xi_{n+1}\left(W_{n+1}^{*T}\Psi_{n+1}\left({\bar{x}}_{n+1}\right)+\varepsilon_{n+1}\right)+\frac{1}{\mu_{n+1}}{\tilde{\theta }}_{n+1}{\dot{\hat{\theta }}}_{n+1}-\xi_{n}\xi_{n+1}-k_{n+1}\xi_{n+1}^{2}\\ & -k_{pn+1}\xi_{n+1}^{p+1}-k_{qn+1}\xi_{n+1}^{q+1}-{\hat{\theta }}_{n+1}\left|\xi_{n+1}\right|\left\|\Psi_{n+1}\left({\bar{x}}_{n+1}\right)\right\| \end{array}$$
when $\left\|W_{n+1}^{*}\right\|=\theta_{n+1}$, we have
(54)$$\xi_{n+1}W_{n+1}^{*T}\Psi_{n+1}\left({\bar{x}}_{n+1}\right)\leq \theta_{n+1}\left|\xi_{n+1}\right|\left\|\Psi_{n+1}\left({\bar{x}}_{n+1}\right)\right\|$$
and
(55)$$\xi_{n+1}\varepsilon_{n+1}\leq \frac{1}{2}\xi_{n+1}^{2}+\frac{1}{2}\varepsilon_{n+1}^{2}$$
then we have
(56)$$\begin{array}{cc} {\dot{V}}_{n+1} & =-{\tilde{\theta }}_{n+1}\left|\xi_{n+1}\right|\left\|\Psi_{n+1}\left({\bar{x}}_{n+1}\right)\right\|+\frac{1}{2}\xi_{n+1}^{2}+\frac{1}{2}\varepsilon_{n+1}^{2}+\frac{1}{\mu_{n+1}}{\tilde{\theta }}_{n+1}{\dot{\hat{\theta }}}_{n+1}\\ & -\xi_{n}\xi_{n+1}-k_{n+1}\xi_{n+1}^{2}-k_{pn+1}\xi_{n+1}^{p+1}-k_{qn+1}\xi_{n+1}^{q+1} \end{array}$$
where
(57)$${\tilde{\theta }}_{n+1}={\hat{\theta }}_{n+1}-\theta_{n+1}$$
choose the NN adaptive law as
(58)$${\dot{\hat{\theta }}}_{n+1}=\mu_{n+1}\left(\left|\xi_{n+1}\right|\left\|\Psi_{n+1}\left({\bar{x}}_{n+1}\right)\right\|-\rho_{pn+1}{\hat{\theta }}_{n+1}^{p}-\rho_{qn+1}{\hat{\theta }}_{n+1}^{q}\right),{\hat{\theta }}_{n+1}\left(0\right)=0$$
where $\mu_{n+1}>0,\rho_{pn+1}>0,\rho_{qn+1}>0$, then we have
(59)$$\begin{array}{cc} {\dot{V}}_{n+1} & =-{\tilde{\theta }}_{n+1}\left|\xi_{n+1}\right|\left\|\Psi_{n+1}\left({\bar{x}}_{n+1}\right)\right\|+\frac{1}{2}\xi_{n+1}^{2}+\frac{1}{2}\varepsilon_{n+1}^{2}\\ & +{\tilde{\theta }}_{n+1}\left(\left|\xi_{n+1}\right|\left\|\Psi_{n+1}\left({\bar{x}}_{n+1}\right)\right\|-\rho_{pn+1}{\hat{\theta }}_{n+1}^{p}-\rho_{qn+1}{\hat{\theta }}_{n+1}^{q}\right)\\ & -\xi_{n}\xi_{n+1}-k_{n+1}\xi_{n+1}^{2}-k_{pn+1}\xi_{n+1}^{p+1}-k_{qn+1}\xi_{n+1}^{q+1} \end{array}$$
based on inequalities from [7], the following hold:(60)$$\begin{array}{l} -\rho_{pn+1}{\tilde{\theta }}_{n+1}{\hat{\theta }}_{n+1}^{p}\leq -\varsigma_{pn+1}{\tilde{\theta }}_{n+1}^{p+1}+\upsilon_{pn+1}\theta_{n+1}^{p+1}\\ -\rho_{qn+1}{\tilde{\theta }}_{n+1}{\hat{\theta }}_{n+1}^{q}\leq -\varsigma_{qn+1}{\tilde{\theta }}_{n+1}^{q+1}+\upsilon_{qn+1}\theta_{n+1}^{q+1} \end{array}$$
where $\rho_{pn+1},\varsigma_{pn+1},\upsilon_{pn+1},\rho_{qn+1},\varsigma_{qn+1},\upsilon_{qn+1}>0$, therefore, we have
(61)$$\begin{array}{cc} {\dot{V}}_{n+1} & \leq -\left(k_{n+1}-\frac{1}{2}\right)\xi_{n+1}^{2}+\frac{1}{2}\varepsilon_{n+1}^{2}-\varsigma_{pn+1}{\tilde{\theta }}_{n+1}^{p+1}+\upsilon_{pn+1}\theta_{n+1}^{p+1}\\ & -\varsigma_{qn+1}{\tilde{\theta }}_{n+1}^{q+1}+\upsilon_{qn+1}\theta_{n+1}^{q+1}-\xi_{n}\xi_{n+1}-k_{pn+1}\xi_{n+1}^{p+1}-k_{qn+1}\xi_{n+1}^{q+1}\\ & \leq -\xi_{n}\xi_{n+1}-k_{pn+1}\xi_{n+1}^{p+1}-k_{qn+1}\xi_{n+1}^{q+1}-\varsigma_{pn+1}{\tilde{\theta }}_{n+1}^{p+1}-\varsigma_{qn+1}{\tilde{\theta }}_{n+1}^{q+1}+\delta_{n+1} \end{array}$$
where
(62)$$\delta_{n+1}=\frac{1}{2}\varepsilon_{n+1}^{2}+\upsilon_{pn+1}\theta_{n+1}^{p+1}+\upsilon_{qn+1}\theta_{n+1}^{q+1}$$

**Theorem** **1.***Consider the non-affine pure-feedback nonlinear system (1), based on the homeomorphism mapping and adaptive fixed-time neural network control scheme, choose the virtual control law as (8), (27), the adaptive fixed-time law (16) as (35), and the actual controller as (47). The tracking error system is practical fixed-time stability, and the upper bound of the settling time* T*is independent of the initial parameters. The settling time*T*satisfies*(63)$$T\leq T_{\max}=\frac{2^{\frac{3-p}{2}}}{k_{p}\left(1-p\right)}+\frac{2}{k_{q}\left(q-1\right)}$$

**Proof.** Select the following Lyapunov function
(64)$$V=\sum_{i=1}^{n+1}V_{i}$$
then it has
(65)$$\dot{V}\leq -\sum_{i=1}^{n+1}\left(k_{pi}\xi_{i}^{p+1}+\varsigma_{pi}{\tilde{\theta }}_{i}^{p+1}\right)-\sum_{i=1}^{n+1}\left(k_{qi}\xi_{i}^{q+1}+\varsigma_{qi}{\tilde{\theta }}_{i}^{q+1}\right)+\sum_{i=1}^{n+1}\delta_{i}$$
Based on Lemma 1
(66)$$\begin{array}{c} \sum_{i=1}^{n+1}\left(k_{pi}\xi_{i}^{p+1}+\varsigma_{pi}{\tilde{\theta }}_{i}^{p+1}\right)\geq k_{p}{\left(\sum_{i=1}^{n+1}\left(\frac{\xi_{i}^{2}}{2}+\frac{1}{2\mu_{i}}{\tilde{\theta }}_{i}^{2}\right)\right)}^{\frac{p+1}{2}}\\ \sum_{i=1}^{n+1}\left(k_{qi}\xi_{i}^{q+1}+\varsigma_{qi}{\tilde{\theta }}_{i}^{q+1}\right)\geq k_{q}{\left(\sum_{i=1}^{n+1}\left(\frac{\xi_{i}^{2}}{2}+\frac{1}{2\mu_{i}}{\tilde{\theta }}_{i}^{2}\right)\right)}^{\frac{p+1}{2}} \end{array}$$
where
(67)$$\begin{array}{l} k_{p}=\min\left(2^{\frac{p+1}{2}}k_{pi},2^{\frac{p+1}{2}}\mu_{i}^{\frac{p+1}{2}}\varsigma_{pi}\right),i=1,2,3\cdots n+1\\ k_{q}=\min\left(2{\left(n+1\right)}^{\frac{1-q}{2}}k_{qi},2{\left(n+1\right)}^{\frac{1-q}{2}}\mu_{i}^{\frac{q+1}{2}}\varsigma_{qi}\right),i=1,2,3\cdots n+1\\ \delta =\sum_{i=1}^{n+1}\delta_{i} \end{array}$$
(68)$$\dot{V}\leq -k_{p}V^{\frac{p+1}{2}}-k_{q}V^{\frac{q+1}{2}}+\delta$$
based on Lemma in [6], the system is practically fixed-time stability. □

**Remark** **2.**
*A new adaptive neural network control strategy is designed. The control objective is to drive the output signal of the error system to track the expected signal in a fixed-time. The neural network is used to approximate the unknown function of the system and design a fixed-time adaptive law to update the weight of the neural network. Without considering the initial conditions, the setting time can be designed by selecting the controller parameters. Based on the fixed-time stability theory, it is proved that the controller can realize the fixed-time stability of the closed-loop system.*


**Remark** **3.**
*The control deviation is obtained from the given value and the actual output value of the system, the fixed-time adaptive laws are designed by the homeomorphic mapping of the deviation, and the neural network weights are trained through the adaptive rate to form the control signal, to change the regulation quality of the system. This forms a fixed-time adaptive neural network control system, and its control structure is shown in Figure 1.*


**Remark** **4.**
*Programming according to the control algorithm described in equation to Equations (4), (23), (43) and the program block diagram is shown in Figure 2*
*Step* *1:**Calculate the control deviation* $z_{i}$*by value and output value.**Step* *2:**Calculate* $\xi_{i}$*according to the principle of homeomorphic mapping.**Step* *3:*
*Design the fixed-time adaptive laws to train the weights of the neural network.*
*Step* *4:*
*Design the neural network to estimate the nonlinear system.*
*Step* *5:**Repeat Step 1 to Step 4 when* i≤n+1.*Step* *6:*
*The control variables are determined based on backstepping control.*



## 4. Numerical Examples

This section gives two examples to show the effectiveness of the proposed control scheme.

A.Mathematical example

The nonlinear dynamics is
(69)$$\begin{array}{l} {\dot{x}}_{1}=x_{2}\sin\left(x_{1}\right)+\left(3+x_{1}^{2}\right)\left(x_{2}+x_{2}^{3}\right)\\ {\dot{x}}_{2}=2x_{1}x_{2}\left(\sin\left(x_{1}\right)+x_{2}^{2}\right)+x_{3}+\frac{x_{3}^{3}}{7}\\ {\dot{x}}_{3}=x_{1}+x_{1}x_{2}+x_{3}+u\\ y=x_{1} \end{array}$$
Consider the system state
(70)$$z_{1}=y-y_{d}$$
Choose the homeomorphism mapping
(71)$$\xi_{1}=\mathrm{arctanh}\left(z_{1}\right)$$
and adaptive functions have the following form:(72)$$\xi_{2}=\xi_{1}+\xi_{1}^{\frac{3}{5}}+\xi_{1}^{\frac{5}{3}}+sign\left(\xi_{1}\right){\hat{\theta }}_{1}\left\|\Psi_{1}\left({\bar{x}}_{2}\right)\right\|$$
and controller has the following form:(73)$$u=\left(1-z_{3}^{2}\right)\left(-\xi_{2}-\xi_{3}-\xi_{3}^{\frac{3}{5}}-\xi_{3}^{\frac{5}{3}}-sign\left(\xi_{3}\right){\hat{\theta }}_{3}\left\|\Psi_{3}\left({\bar{x}}_{3}\right)\right\|\right)$$
where $y_{d}=\sin\left(t\right)$ being the desired signal. Select the initial parameters as $x={\left(1,0,0\right)}^{T}$, and the neural network parameters chosen zeros.

The simulation results are shown in Figure 3, Figure 4, Figure 5 and Figure 6. Figure 3 depicts the tracking curve of the given value and output value. It can be seen from the figure that the tracking error can be sufficiently small in fixed-time and the system output is bounded. Figure 4 shows that the system state is bounded and can converge to zero in fixed time. Figure 5 depicts the tracking errors’ tracking curve, which shows that the tracking errors are bounded. Because $\tanh\left(\xi_{i}\right)=z_{i},i=1,2,3$, therefore, the system states $z_{i},i=1,2,3$ are bounded with $\left|z_{i}\right|<1$. Figure 6 shows the time response of the output, the output is bounded, and its value is constant after a fixed time.

B.Robot model

Consider a robot model [31] is
(74)$$M_{r}{\ddot{q}}_{r}+\frac{1}{2}m_{r}gl_{r}\sin\left(q_{r}\right)=\tau_{r}$$
where $q_{r}$ is angle displacement, g and $M_{r}$ are the gravitational acceleration and moment of inertia, respectively, and $m_{r}$ is the mass of link and $l_{r}$ represents its length, $\tau_{r}$ is the considered input torque. If $x_{1}=q_{r},x_{2}={\dot{q}}_{r}$, and $u=\tau_{r}$, the dynamic system can be transformed as follows:(75)$$\left\{\begin{array}{l} {\dot{x}}_{1}=x_{2}\\ {\dot{x}}_{2}=-\frac{mgl_{r}}{2M_{r}}\sin\left(x_{1}\right)+\frac{1}{M_{r}}u \end{array}\right.$$

For simulation process, the neural networks adaptive fixed-time control, the $y_{d}=0.1\sin\left(t\right)$ being the desired signal.

The simulation results are shown in Figure 7 and Figure 8. Figure 7 depicts the tracking curve of the given value and output value. It can be seen from the figure that the tracking error can be sufficiently small in fixed-time and the system output is bounded. Figure 8 shows the time response of the control input.

## 5. Conclusions

So far, great breakthroughs have been made in the research of adaptive neural network tracking controls for nonlinear systems, but there are still some control problems to be solved. In this paper, a new fixed-time adaptive neural network tracking control strategy is designed for pure-feedback non-affine nonlinear constrained systems. Based on the backstepping control technology, the fixed-time adaptive neural network function of the error system is designed. The setting time by the control parameters and adaptive law gain parameters, that is, the control performance can be guaranteed without initial conditions, which is more practical than the control algorithm based on Lyapunov stability theory.

## Figures and Tables

**Figure 1 entropy-24-00737-f001:**
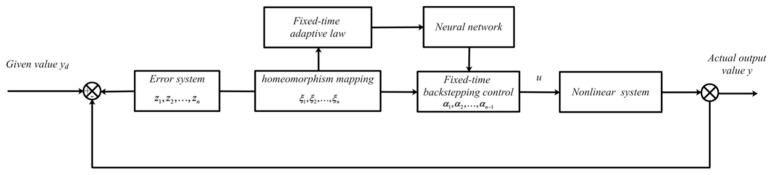
Fixed-time adaptive neural network control system.

**Figure 2 entropy-24-00737-f002:**
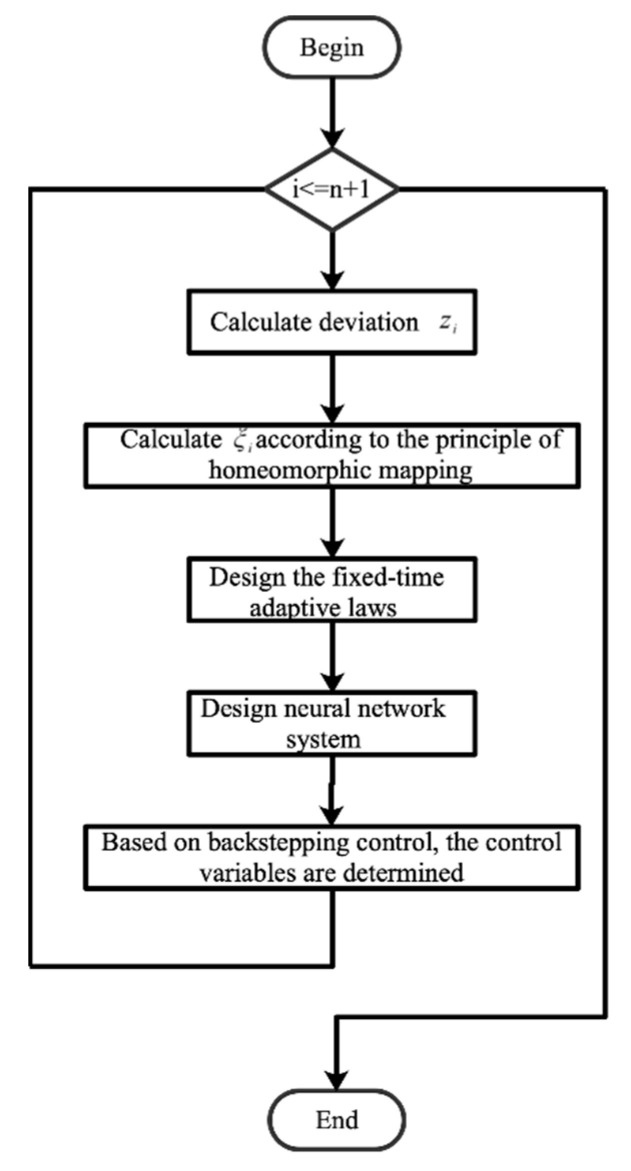
Fixed-time adaptive neural network control algorithm.

**Figure 3 entropy-24-00737-f003:**
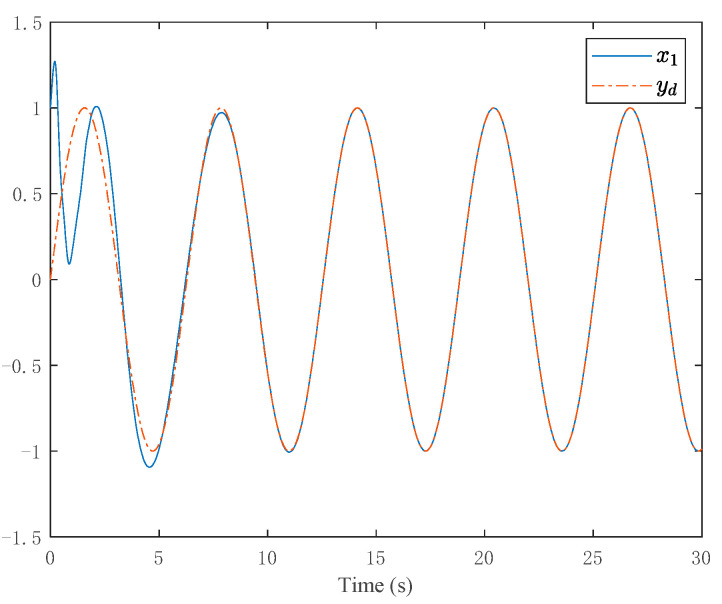
Trajectories of the output and the desired signal.

**Figure 4 entropy-24-00737-f004:**
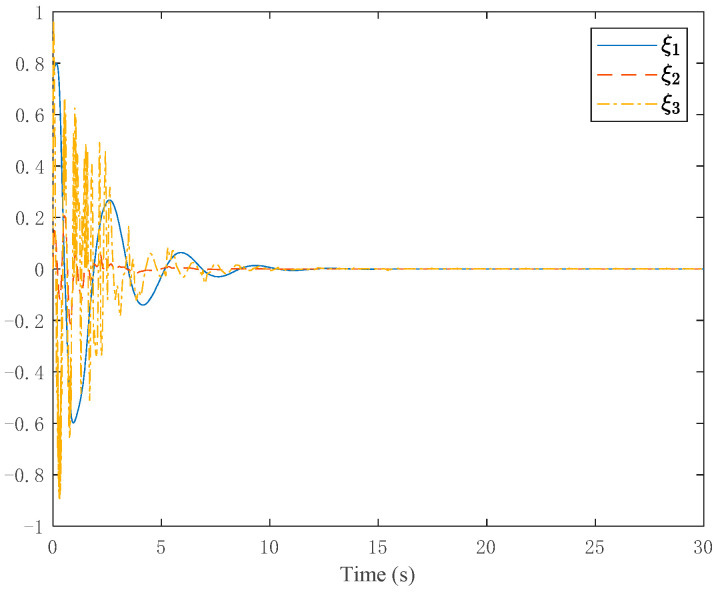
Trajectories of the homeomorphism mapping states.

**Figure 5 entropy-24-00737-f005:**
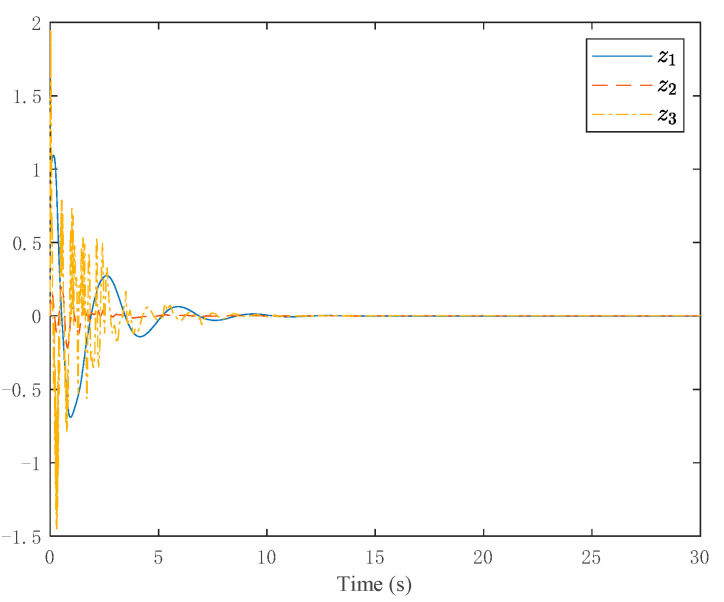
Trajectories of the system states.

**Figure 6 entropy-24-00737-f006:**
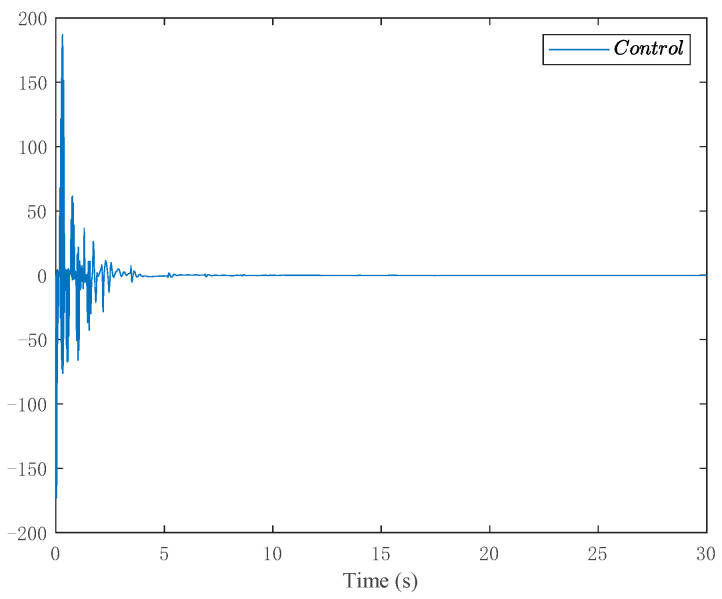
Trajectories of the controller.

**Figure 7 entropy-24-00737-f007:**
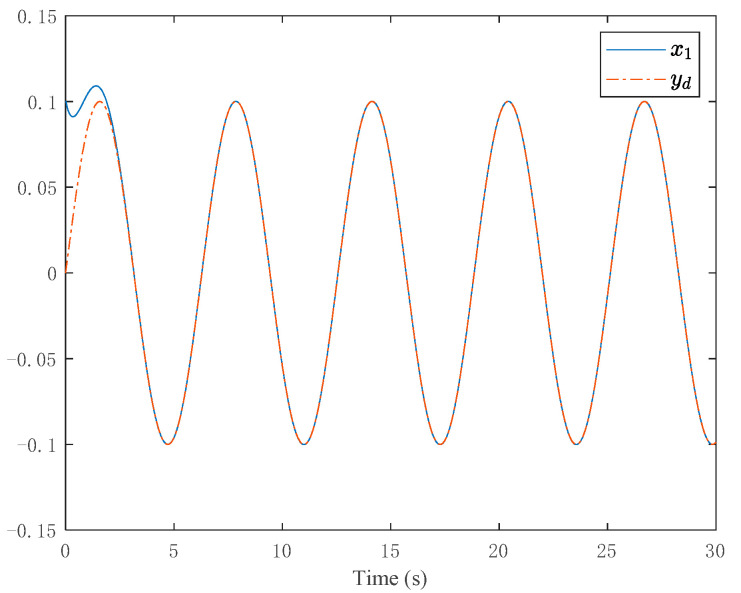
Trajectories of the output and the desired signal.

**Figure 8 entropy-24-00737-f008:**
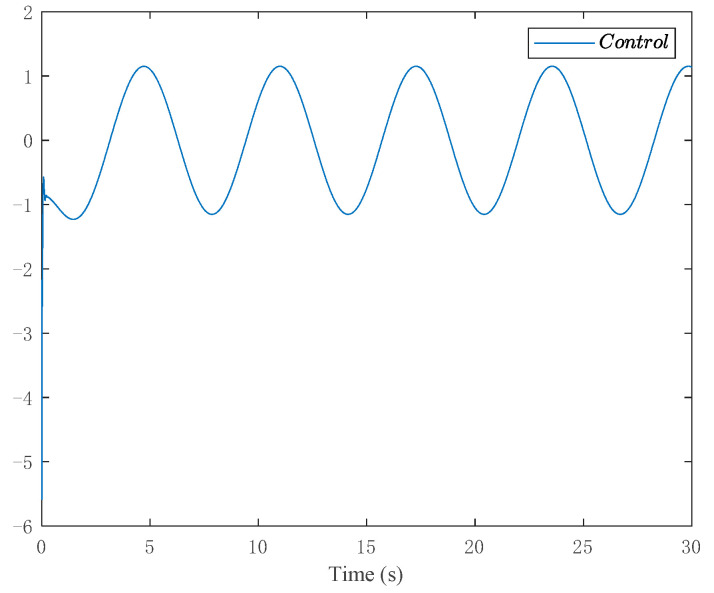
Trajectories of the controller.

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
