# Peer review of "Adaptive Fixed-Time Neural Networks Control for Pure-Feedback Non-Affine Nonlinear Systems with State Constraints"

_entropy, 2022, doi:10.3390/e24050737_

Round 1

Reviewer 1 Report

What are the novelties for the proposed Lyapunov stability analysis and the fixed time stability analysis?

(28) is not immediately clear. Please explain this and also (31), which is related to the argument.

It would be useful if the derivation of (60) and (61) are verified in the proof. Some details should be given.

What are the time complexity and space complexity for the tixed-time adaptive neural network control algorithm?

The numerical example is minimum. You should consider a larger scale system which could verify the theoretical results. A realistic application is also needed given the purpose of the paper stated in the introduction.

The writing should be improved in general.

Author Response

Dear Editor,

Paper: Entropy- 1697987.v1

Adaptive Fixed-Time Neural Networks Control for Pure-Feedback Non-Affine Nonlinear Systems with State Constraints

Many thanks for your instruction for the resubmission of this study. We have revised the paper in line with the editor and reviewer recommendations/corrections and positively responded to all of the comments/questions raised. The replies to the editor and the reviewers are attached to the cover letter.

In addition, we have shown our gratitude to the editor and the anonymous reviewers for their helpful comments and constructive suggestions about the revision of the paper in the acknowledgement.

Should you have any query regarding this paper, please contact me again.

Yours sincerely,

Jianhua Zhang

Reviewer 2 Report

This paper proposed a fixed-time adaptive neural network control strategy for the pure feedback non-affine nonlinear systems with state constraints. Here are the main concerns of the reviewer:
1. Although this paper claims to address the state constraint in the abstract and several places, it is not clear how the control input considers it. Particularly, in the simulation results, there are no discussions about the state constraint. 
2. What is the accuracy of the trained neural network? Will the training and testing error degrade the system performance?
3. The authors need added comparison with other controllers, such as PID, etc., to show the benefits of the proposed strategy.
4. The presentation of Section 3 needs to be significantly improved. The current proof and derivation process is hard to follow. Many symbols are not well defined and confusing. For instance, in Equ.(4), f_1(x_1,x_2) is expressed as a function of x_1 and x_2, while in Equ.(1), it is only determined by x_2. In Equ.(7), what is \theta?

Author Response

(The authors gave the same response as above.)

Round 2

Reviewer 1 Report

I have no further comments. The paper can be accepted. 

Reviewer 2 Report

No further comments.